# Research on the Influence of Bed Joint Reinforcement on Strength and Deformability of Masonry Shear Walls

**DOI:** 10.3390/ma12162543

**Published:** 2019-08-09

**Authors:** Radosław Jasiński

**Affiliations:** Department of Building Structures, Silesian University of Technology, ul. Akademicka 5, 44-100 Gliwice, Poland; radoslaw.jasinski@polsl.pl; Tel.: +48-32-237-11-27

**Keywords:** masonry structures, shear walls, Clay Brick (CB), calcium-silicate (Ca-Si) masonry units, Autoclaved Aerated Concrete (AAC) masonry units, bed joint reinforcement, shear strength, strain angle, wall stiffness

## Abstract

The areas of Central and Eastern Europe and, thus, Poland are not exposed to the effects of seismic actions. Any possible tremors can be caused by coal or copper mining. Wind, rheological effects, the impact of other objects, or a nonuniform substrate are the predominant types of loading included in the calculations for stiffening walls. The majority of buildings in Poland, as in most other European countries, are low, medium-high brick buildings. Some traditional materials, like solid brick (>10% of construction materials market) are still used, but autoclaved aerated concrete (AAC) and cement-sand calcium-silicate (Ca-Si) elements with thin joints are prevailing (>70% of the market) on the Polish market. Adding reinforcement only to bed joints in a wall is a satisfactory solution (in addition to confining) for seismic actions occurring in Poland that improves ULS (ultimate limit state) and SLS (serviceability limit state). This paper presents results from our own tests on testing horizontal shear walls without reinforcement and with different types of reinforcement. This discussion includes 51 walls made of solid brick (CB) reinforced with steel bars and steel trusses and results from tests on 15 walls made of calcium-silicate (Ca-Si) and AAC masonry units reinforced with steel trusses and plastic meshes. Taking into account our own tests and those conducted by other authors, empirical relationships were determined on the basis of more than 90 walls. They are applicable to the design and construction phases to determine the likely effect of reinforcements on cracking stress that damage shear deformation and wall stiffness.

## 1. Introduction

Testing the impact of reinforcement on masonry shear walls dates back to the turn of the 1940s and 1950s, nearly 80 years after the first documented tests on unreinforced wall shearing performed by Bauschinger in 1873 [1] and 130 years after the first test on reinforced walls subjected to eccentric compressive loads [2]. The aims of tests on wall shearing that have been conducting until now are the verification of models adapted from concrete structures [3,4,5,6,7], practical estimation of the effects of different types of (metallic and nonmetallic) reinforcement [8,9], and masonry units and mortar types on important mechanical parameters of the masonry (crack resistance, load capacity, and deformability). Considering seismic actions, the reinforcement in stiffening walls is placed in bed joints and vertical cores [10,11,12,13] and nonmetallic reinforcements placed in a similar way are also employed [14,15,16]. Tests are conducted on single walls or whole buildings [17,18,19,20,21,22,23]. There are also tests of confined walls or of the influence of different types of reinforcements [24,25,26]. Seismic actions observed in Poland and in the area of Central and Eastern Europe are connected with industrial anthropogenic causes—mining tremors and geological and geotechnical actions—and they are definitely different from typical effects of earthquakes. Therefore, reinforcement in masonry structures is intended to increase crack resistance, shear strength, and wall stiffness and is only applied in bed joints in the masonry. There are still few documented tests on such types of reinforcement [27,28,29,30,31]. The need for tests is usually accompanied by the tendency to reduce the wall thickness and meeting ULS (ultimate limit state) and SLS (serviceability limit state) criteria according to current design standards PN-EN 1996-1-1 [32]. The aim of these tests was to determine the effect of different types of reinforcement on bed joints, of which the number was reduced to the minimum. Tests were done in accordance with the standard [32]. Conducted analyses were to provide approximate empirical relations to determine the effect of the applied reinforcement on strength parameters and stiffness of the wall.

## 2. Research Programme

The previous tests on shearing of reinforced masonry walls were characterised by a wide diversity that substantially reduced the possibility for comparing results from analyses and for drawing practical conclusions. Experiments were performed on units of different shapes and dimensions at test stands used for various static schemes. Models were loaded cyclically, occasionally, or temporarily. Moreover, research models were made of materials and different types of inserts or reinforcement arrangements not used in Poland, and the number of element series was rather low.

To avoid the above weaknesses, the following aims about our own tests were made:to use the most common materials in Poland to erect masonry structures,to use the minimum amount of reinforcement,to use squat walls with an h/l ratio close to real structures,to build a unique test stand to perform tests on shearing and compression at the same time in a partially fixed static scheme.

The following materials were used to make models in accordance with the above assumptions:ceramic solid brick (CB), calcium-silicate masonry units (Ca-Si) from group I, and AAC masonry units from 600 density class,cement-lime mortar with a cement:lime:sand ratio of 1:1:6 to make CB walls and the system mortar for thin joints for Ca-Si and AAC walls,two types of reinforcement for bed joints in walls made of solid brick: smooth rebars with a diameter of 6 mm and made of stainless steel and structural reinforcement in the form of steel, galvanized trusses, in which the strips were made of steel rebars with a diameter of 5 mm and the struts were made of rebars with a diameter of 3.75 mm as in Figure 1a,plastic meshes and steel trusses for thin joints as in Figure 1b.

Units from solid brick were reinforced with smooth bars made from stainless steel 1H18N95T-1.4541 and had a diameter of 6 mm (series: HC-ZPI and HC-ZPII), and that for the structural reinforcement (series: HC-ZKI and HC-ZKII) welded galvanized trusses of type MURFOR^®^RND/Z/200 (NV Bekaert SA, Zwevegem, Belgium). Those types of reinforcement were selected due to practical reasons. The reinforcement in the form of unbounded rebars, although not recommended by the standard [33], is the simplest type of reinforcement most commonly used for strengthening cracked walls. Moreover, that type of reinforcement was unlikely to cause any negative effects according to a few tests taken on walls, including horizontal shear walls [17,29]. One of currently recommended by the standard [33] types of structural reinforcement—welded trusses—was accepted for tests. The reinforcement in the form of steel trusses of type MURFOR^®^EFS/Z/140 intended for thin joints was used in models made of silicate masonry units. Steel trusses were composed of two stripes made of steel flat bars (8 × 1.5 mm) joined with diagonal struts with a diameter of 1.5 mm. Plastic meshes towards the wall length were composed of weft fibres, and that towards the wall thickness were composed of warp fibres. A waft comprised two strands; each of them had a cross section similar to a circle having a diameter of 0.3 mm, whereas a weft was formed from a single strand with a cross section similar to a rectangle with dimensions of 1.5 mm × 0.22 mm. Percentage contents of the reinforcement in brick models were *ρ* = 0.05% and *ρ* = 0.1% and in walls made of silicate and AAC masonry units was *ρ* = 0.07%. In the case of solid brick walls, the proportions of the wall dimensions were *h/l* ≈ 0.84, and for walls made of calcium silicate and AAC masonry units, those proportions were *h/l* ≈ 0.55.

### 2.1. Masonry Walls Made of Clay Brick

Five series of units were prepared for testing brick walls; see Table 1. At the beginning, units series were prepared without reinforcement (HC), reinforced with bars (HC-ZPI, in every sixth bed joint; and HC-ZPII, in every third bed joint), and reinforced with trusses (HC-ZKI, in every sixth bed joint; and HC-ZKII, in every third bed joint).

All models had the same shape and dimensions: length *l* = 1.68 m, height *h* = 1.415 m (*h/l* = 0.84), and thickness *t* = 0.25 m. Units reinforced with rebars (Figure 1a) and trusses (Figure 1b) (series HC-ZPI and HC-ZKI) had two smooth bars with a diameter of *ϕ*6 mm placed in every sixth bed joint (*s* = 450 mm) which contributed to the reinforcement percentage (*ρ* = 0.05%); see Figure 2a. In the units of series HC-ZPII and HC-ZKII, the reinforcement percentage was doubled to *ρ* = 0.10% by introducing rebars in every third joint (*s* = 450 mm); see Figure 2b.

The walls were made from solid brick having normalised compressive strength *f_b_* = 28.8 N/mm^2^ (acc. to PN-EN 772-1 [34]) with cement–lime mortar having compressive strength *f_m_* = 9.67 N/mm^2^ (acc. to EN 1015-11:2001 [35]). Average compressive strength of masonry determined according to EN 1052-1 [36] was *f_c,m_* = 8.17 N/mm^2^, and the modulus of elasticity was *E_cm_* = 3110 N/mm^2^ according to Reference [36]. The initial shear value determined according to PN-EN 1052-3 [37] was *f_v,o_* = 0.452 N/mm^2^. The average yield strength of smooth stainless-steel bars was *f_y_* = 592 N/mm^2^ and of strips and struts in steel trusses were *f_y_* = 701 N/mm^2^ (strips of 5 mm diameter) and *f_y_* = 625 N/mm^2^ (bracking of 3.75 mm diameter), respectively.

### 2.2. Masonry Walls Made of Calcium-Silicate (Ca-Si) Masonry Units

Tests were performed on walls with a height/length ratio equal to *h/l* = 2.45/4.50 (Figure 3) and thickness of 0.18 m, divided into three series (HOS, HOS-Z1-S, and HOS-Z2-S). The walls were reinforced with galvanised steel trusses (Figure 1c) and plastic meshes (Figure 1d) placed in each bed joint to provide the reinforcement percentage of 0.07%. Tests were conducted on seven units in total. The research programme for the walls is shown in Table 2.

Walls were made from silicate masonry units (240 × 220 × 180 mm) having compressive strength *f_b_* = 17.7 N/mm^2^ (acc. to the standard [34]) with system mortar for thin joints having compressive strength *f_m_* = 18.20 N/mm^2^ (acc. to the standard [35]). Head joints were not filled in models. The compressive strength of masonry determined according to the standard [36] was *f_c,mv_* = 11.3 N/mm^2^, and the modulus of elasticity was *E_cm_* = 7833 N/mm^2^. The initial shear value determined according to the standard [37] was *f_v,o_* = 0.70 N/mm^2^. The average yield strengths of strips and struts in steel trusses were *f_y_* = 685 N/mm^2^ (flat bars 1.5 × 8 mm) and *f_y_* = 821 N/mm^2^ (bracking of 1.5 mm diameter), respectively. 

### 2.3. Masonry Walls Made of Autoclaved Aerated Concrete (AAC) Masonry Units

The walls were made of autoclaved aerated concrete masonry units with a height/length ratio *h/l* = 2.43/4.43, (Figure 4) and thickness of 0.18 m, divided into three series: HOS-AAC, HOS-Z1-AAC, and HOS-AAC. As in the case of models made of calcium silicate masonry units, the applied reinforcement was in the form of galvanized steel trusses (Figure 1c) and plastic meshes (Figure 1d) placed in every bed joint. The achieved reinforcement percentage was 0.07%. There was no reinforcement in other walls. Tests were performed on eight units in total. The research programme for the walls is shown in Table 3.

Models were made from AAC units (695 × 240 × 180 mm) having compressive strength *f_b_* = 3.65 N/mm^2^ [34] with system mortar for thin joints having compressive strength *f_m_* = 6.10 N/mm^2^ [35]. Head joints were not filled in models. The compressive strength of masonry was *f_c,mv_* = 2.97 N/mm^2^ [36], and the modulus of elasticity was *E_cm_* = 2041 N/mm^2^. The initial shear value was *f_vo_* = 0.31 N/mm^2^ [37]. Models were reinforced with steel trusses and plastic meshes having the same parameters as in the calcium silicate masonry units.

## 3. Test Stand and Testing Technique

The author designed and performed a test stand for testing horizontal shear walls (Figure 5a). It consisted of two steel columns *4* and *5*; three horizontal spandrel beams *2, 3,* and *7*; and four sets of tendons *6* to induce compressive prestress *σ_c_*. Both columns differed in shape and the method of fixing to the laboratory strong floor. Column *5* had a closed box section (2 × I 500), rigidly fixed with four screws (ø65 mm). Actuator *8* with the force of 3000 kN was fixed to the upper part of the column in a way ensuring the smooth change of its position. At the bottom part, there was a horizontally articulated support for spandrel beam *3*. Two steel knee braces *11* were articulated with the column and the resistor fixed to the strong floor. Knee braces were used to neutralise the effect of column bending. Column *4* had a two-branch section (2 × [260). Their branches were joined in the upper part with a lacing, and in the bottom part, they were welded to the slab of the laboratory strong floor with two screws (ø65 mm). The openings were made in the column branches to stabilize the movable horizontal “crossbar” *10* (2 × I 300) using the vertical support (with the dynamometer). Horizontal spandrel beams *2*, *3*, and *7* also had different shapes and purposes. Spandrel beams *2* and *3* had closed box sections. Between their stripes, bars with a diameter of 20 mm were welded across (to assure the adhesion of the monolithic concrete). During tests, the support at column *5* precluded the horizontal movement of spandrel beam *3*, and its vertical movement was precluded by support *12*. Spandrel beam *2* placed in the upper part of the model surface was horizontally sliding, supported vertically on the bearer in column *4*. At spandrel beam edge *2*, there was a cylindrical joint responsible for transmitting horizontal load from spandrel beam *7* (through the steel pin ø100 mm) to the testing unit. Branches of spandrel beam *7* were made of channels [260 and closed transversely with the system 2 × I 300. From column side *5* within the longitudinal axis of actuator *8*, spandrel beam *7* was equipped with dynamometer *9* with the working range of 3000 kN, used to transmit horizontal shearing forces.

Compressive stress was induced by eight-tendon “compressive” systems *6*. A single system was composed of three small, horizontal “spandrel beams”—the upper one placed on the main spandrel beam *2* and two bottom ones placed under spandrel beam *3*. Small, horizontal ”spandrel beams” were joined with steel tendons (ø45 mm). Each set had a dynamometer with the measurement range of 250 kN and a fixing screw (to maintain tendon tension which was initially induced by two actuators). To maintain the constant value *σ*_c_ during tests, both tendons from the set were equipped with a compensation spring to minimise the impact of steel relaxation in tensioned tendons and vertical deformations of the wall. When the test stand was set, each model was tested in a sequential mode (Figure 5c). The first part of the tests consisted of exerting compressive prestress *σ_c_*(*N_c_*) on test units perpendicularly to the plane of bed joints with tendon systems. In the second part, units under compressive prestress were loaded with horizontal force *H*. The loading programme for all models included three cycles of loading and unloading. The load of 10 kN, that is ca. 5% of predicted failure load *H_u_*, was exerted in two first cycles. At that time, readings from measuring instruments were controlled and mobile elements of the stand were adjusted to the starting positions. In the third failure cycle, models were temporarily loaded every time until the force increase was not recorded on dynamometer *9* and, simultaneously, an increase in horizontal displacements of spandrel beams *2* and *7* was observed. Loading was changed progressively by 10 kN every 2 minute, and readings from inductive displacement transducers in dynamometers were recorded with the automatic measurement stand. Forces generated by tendon systems *6* were measured with the dynamometer Utilcell 750 with an operating capacity of 250 kN and reading accuracy of 0.1 kN, whereas the horizontal force *H* and support reaction *R*_A_ were measured with electro-resistant dynamometers CT 300 and CT 100, having an operating capacity of 3000 kN and 1000 kN, respectively, and an accuracy of 0.5 kN. A frame structure was used to measure shear strain and deformation angles. The structure was fixed to both sides of each test model. The frame measurement system was fixed in the central part of the model made of a solid brick wall and covered the substantial area of the wall made from silicate or AAC masonry units. The measurement system was firmly fixed to the wall surface using an epoxy adhesive. The system was symmetrically fixed to both sides of the test model such that the diagonal centre of the test model corresponded to the diagonal centre of the frame system; see Figure 6. Displacements Δ*_c_*, Δ*_f_*, Δ*_i_*, Δ*_j_*, Δ*_g_*, and Δ*_h_* were measured along each of four sides (*c*, *f*, *i*, and *j*) of the test model and along two diagonals (g and h) of the frame system, using inductive converters of displacement with the accuracy of 0.002 mm. The range of indications was ± 10 mm. Changes in the lengths of sides *l_cc_**, l_fc_**, l_ic_**,* and *l_jc_* (at the *i*th level of loading) and diagonals *l_gc_* and *l_hc_* and partial angles of shear strain Θ*_j_* (*j* = 1, 2, 3, 4), theoretically separated from the deformed measurement system, were determined.

The terms global angle of shear strain or global angle of shear deformation at the phase after cracking were used to describe the wall behaviour under horizontal shearing. Partial values of the global angle of shear strain Θ*_j_* were determined according to the law of cosines, on the basis of average changes in the length of measuring bases (Figure 6b).

A triangle with *l_fc_*, *l_hc_*, *l_jc_* in Equation (1): (1)$$\begin{array}{l} l_{hc}^{2}=l_{fc}^{2}+l_{jc}^{2}-2l_{fc}l_{jc}\cos\left(\frac{\pi }{2}+\Theta_{1}\right)\to \Theta_{1}=-\frac{\pi }{2}+\arccos\left(\frac{l_{fc}^{2}+l_{jc}^{2}-l_{hc}^{2}}{2l_{fc}l_{jc}}\right),\\ l_{gc}^{2}=l_{fc}^{2}+l_{ic}^{2}-2l_{fc}l_{ic}\cos\left(\frac{\pi }{2}-\Theta_{2}\right)\to \Theta_{2}=\frac{\pi }{2}-\arccos\left(\frac{l_{fc}^{2}+l_{ic}^{2}-l_{gc}^{2}}{2l_{fc}l_{ic}}\right),\\ l_{gc}^{2}=l_{jc}^{2}+l_{cc}^{2}-2l_{jc}l_{cc}\cos\left(\frac{\pi }{2}-\Theta_{3}\right)\to \Theta_{3}=\frac{\pi }{2}-\arccos\left(\frac{l_{jc}^{2}+l_{cc}^{2}-l_{gc}^{2}}{2l_{jc}l_{cc}}\right),\\ l_{hc}^{2}=l_{ic}^{2}+l_{cc}^{2}-2l_{ic}l_{cc}\cos\left(\frac{\pi }{2}+\Theta_{4}\right)\to \Theta_{4}=-\frac{\pi }{2}+\arccos\left(\frac{l_{ic}^{2}+l_{cc}^{2}-l_{hc}^{2}}{2l_{ic}l_{cc}}\right). \end{array}$$

The length of measuring bases under the load was determined from the following relationships:  *l_ic_* = *l* + Δ*_i_*,        *l_fc_* = *l_f_*_0_ + Δ*_f_*,  *l_jc_* = *l_j_*_0_ + Δ*_j_*,      *l_gc_* = *l_g_*_0_ + Δ*_g_*,*l_cc_* = *l_c_*_0_ + Δ*_c_*,      *l_hc_* = *l_h_*_0_ + Δ*_h_*.

The average value of the global angle of shear strain of the masonry Θ, (at the *i*th level of loading) was determined as the mean arithmetic value of partial values Θ*_j_* in Equation (2):(2)$$\Theta =\frac{1}{n}\sum_{j=1}^{n=4}\left|\Theta_{j}\right|.$$

A statistical analysis was taken while determining the average global angle of shear strain to verify if extreme (minimum and maximum) values of partial angles of the global angel of shear deformation belong to a specific general population. The population was regarded as four values of shear strain angle Θ*_j_* (*j* = 1, 2, 3, 4) determined at the *i*th level of loading. The assumed critical statistical value of the questioned extreme value was *t*_4,0.05_ = 1.46 after four observations in the specimen and at the significance level equal to 0.05. Maximum and minimum values of shear strain angles were found from Equations (3) and (4):(3)$$\Theta_{\max,i}=\Theta_{i}+t_{4,0.05}S,$$
(4)$$\Theta_{\min,i}=\Theta_{i}-t_{4,0.05}S,$$
where Θ*_i_* is the average value of shear strain angle at the *i*th level of loading and *S* is the standard deviation determined from (Θ*_j_*) shear strain angles at the *i*th level of loading.

Shear stress *τ_i_* was determined as the ratio of horizontal loading *H_i_* and the horizontal area of the masonry in accordance with Equation (5):(5)$$\tau_{i}=\frac{H_{i}}{A_{h}}.$$

General stiffness of the wall *K_i_* was the ratio of the applied force *H_i_* and the corresponding horizontal displacement *u_i_* according to Equation (6):(6)$$K_{i}=\frac{H_{i}}{u_{i}}=\frac{\tau_{i}}{\Theta_{i}}\frac{A_{h}}{h}.$$

The wall stiffness *K_cr_* was determined at the time of observing first cracks*,* and the initial stiffness *K_o_* was determined at the initial phase of loading under stresses 0 < *τ* ≤ 0.05 *τ_u_*. The measured force, at which the first crack was observed in the masonry units or mortar, was considered as the cracking force *_Hcr_*. The corresponding stresses *τ_cr_* were defined as cracking stresses, and the angle Θ*_cr_* was defined as the shear strain angle at the time of cracking. The width of 0.1 mm was regarded as the minimum width of the crack, neglecting all previously observed micro-cracks. A detailed list of existing and visible cracks in the masonry units was prepared to avoid any wrong interpretations of visible cracks. The measured force, at which the model was destroyed, was considered as the destructive force *H_u_*—an increase in force was not further recorded at increasing displacements. Stresses at the top edge of the wall, corresponding to the force *H_u_*, were regarded as ultimate stresses *τ_cr_*, and the angle Θ*_u_* was regarded as the shear deformation angle. 

## 4. Test Results

### 4.1. Morphology of Cracks in Walls

#### 4.1.1. Solid Brick Walls

The crack arrangement was consistent with the course of main tensile stresses of the loaded unit. Cracks in all models developed in the central part in a diagonal direction towards opposite corners of the masonry. The loss of load capacity and the nature of cracks clearly depended on values initial compressive stress. In the units from shear series *σ_c_* = 0, the loss of load capacity was observed at the interface of bricks and mortar. Places where bonded masonry units were present were also cracked. A rapid loss in load capacity was observed at a specific length of the “stepped” cracking—considerably shorter than the diagonal length—see Figure 7a. A similar type of cracks was found in shear units at *σ_c_* = 0.5 N/mm^2^, and units exposed to shearing at *σ_c_* = 1.0 N/mm^2^ and 1.5 N/mm^2^ had cracks in masonry units and mortar layers. No loss in adhesion between masonry units and mortar was observed. 

As the load applied to the central zone was increasing, much more cracks were developed, which were running towards the wall corners. A diagonal crack at the failure phase covered nearly the whole diagonal of the masonry. Significant friction in cracks developed in the central area and caused chipping and spalling of small fragments of the masonry. Vertical (local) cracks or chips were observed almost simultaneously in corner zones, that is, places on which vertical reactions *R_A_* and *R_B_* took place, as the result of considerable compression. No significant difference was noticed in the cracking of walls unreinforced or reinforced with bars; see Figure 7b,c. A little higher number of cracks having a width of ca. 0.1–0.3 mm was only observed in the central part of shear masonry units at *σ_c_* = 1.5 N/mm^2^. Prior to the failure of reinforced units, gentle drops in forces read at the dynamometer and greater width of cracks were observed. No significant changes with reference to models reinforced with bars were observed in walls reinforced with trusses; see Figure 7d,e.

#### 4.1.2. Walls Made of Silicate Masonry Units

At the time of failure, there was one diagonal crack running through bed and head joints (Figure 8a). In the central part of the wall, it joined the horizontal crack in the bed joint. At the same time, a few masonry units in the top layer were crushed. In the unreinforced wall under maximum shear and compressive stress, diagonal cracks were running along the whole height of the wall, that is, through head joints and masonry units (Figure 8b). 

At the time of failure, individual masonry units in the top layer of the wall were crushed. Differences in cracking of reinforced walls were rather subtle. They became clear at the time of failure. Diagonal cracks running through bed and head joints, locally crushed masonry units (Figure 8c,e) with simultaneously broken reinforcement (Figure 8g) were observed in walls reinforced with truss and plastic mesh, sheared at minimum values of compressive stress. No horizontal crack was observed in the bed joint in the central part of the wall. A similar situation took place in shear-reinforced units under compression stress of 1.5 N/mm^2^ (Figure 8c,f). An intensive cracking of masonry units and simultaneous crushing of masonry units was found in the top layer, and the reinforcement was broken in the zone of the greatest damage (Figure 8h). Instead of two diagonal cracks running from support of unreinforced walls, only one diagonal crack was observed in reinforced walls. Also, the top layers of masonry units were crushed.

#### 4.1.3. Walls Made of AAC Masonry Units

One diagonal crack (Figure 9a) running along the wall diagonal was dominating at failure. The unreinforced wall compressed to 1.0 N/mm^2^ had diagonal cracks at the moment of failure. However, vertical cracks predominated at the extension of head joints in masonry units (Figure 9b). 

At the time of failure, individual masonry units in the top layer of the wall were crushed. Like in walls made of calcium-silicate masonry units, no significant differences were observed in the method of cracking. Diagonal cracks running through bed and head joints developed in walls reinforced with truss and plastic mesh, sheared at compressive prestress equal to 0.1 N/mm^2^ (Figure 9c,e). In this case, diagonal cracks did not run along the whole wall diagonal. Rapid breaking of reinforcement was observed at failure (Figure 9g). In reinforced elements subjected to compressive prestress equal to 1.0 N/mm^2^ (Figure 9c,f), cracking was much more intensive than in case of units under minimum compression. In that case, diagonal cracks were predominating and individual masonry units were crushed. Also, the breaking of reinforcements was observed (Figure 9h). The intensity of cracking in reinforced walls at the time of failure was considerably greater when compared to unreinforced walls.

### 4.2. Effect of Reinforcement

#### 4.2.1. Solid Brick Walls

Until the time of cracking of unreinforced units solid brick units with reinforcement, the stress–shear strain angle relationship was nearly directly proportional but clearly depended on values of initial compressive stress. A significant increase in the shear–strain angle was observed in unreinforced walls at the minimum increase in loading (Figure 10). Such an occurrence was not recorded in reinforced units regardless of reinforcement type and percentage, in which the angle value was decreasing after cracking. The comparison of average results from testing two or three units is shown in Table 4.

The greatest impact of the reinforcement on cracking stress was found for shear units. At *ρ* = 0.05%, stress increased by 30% in models reinforced with bars and by 115% in models reinforced with trusses when compared to models of series HC. When the reinforcement percentage was doubled in models under shear stress *σ_c_* = 0, stress increased by 40% for reinforcement with bars and by 120% for reinforcement with trusses in comparison to unreinforced walls. With increasing values *σ_c_*, the impact of reinforcement was clearly weakening in comparison to unreinforced walls. Only in walls with truss reinforcement, a 40% increase in stress values was observed at both levels of the reinforcement percentage. The biggest impact of the reinforcement at the time of failure was only observed in shear units. In comparison to unreinforced units, *τ_u,mv_* increased to 100% (*ρ* = 0.05%) and 110% (*ρ* = 0.10%) in walls reinforced with trusses and to 45% (*ρ* = 0.05%) and 40% (*ρ* = 0.10%) in walls reinforced with bars. When values *σ_c_* were increasing, the reinforcement impact was decreasing proportionally. An increase of 3% (*ρ* = 0.05%) and 20% (*ρ* = 0.10%) was found in masonry units with rebars under maximum compression and of 30% in walls with truss reinforcement at both types of reinforcement percentage. The results for shear stress at the time of cracking and failure as a function of initial compressive stresses are shown in Figure 11a.

Like in case of shear stress, angles of shear strain Θ*_cr,mv_* determined at the time of cracking was increasing proportionally to an increase of initial compressive stress (*σ_c_*). For reinforced walls, those values were lower than values of shear strain angle Θ*_cr,mv_* obtained for unreinforced walls. The greatest decrease in angles was found in shear units in walls reinforced with bars at *σ_c_* = 0 by 50% and in walls with truss reinforcement by 30% (*ρ* = 0.05%) and 40% (*ρ* = 0.10%). Also, shear deformation of walls determined at the time of failure Θ*_u,mv_* increased with increasing compressive stresses. When compared to unreinforced units, values of Θ*_u,mv_* determined for reinforced walls were lower at *σ_c_* = 0 by 50–60% in case of rebar reinforcement and by 40/50% in the case of truss reinforcement. At the highest values *σ_c_* = 1.5 N/mm^2^, shear deformations of walls with truss reinforcement were greater by 8% (*ρ* = 0.10%), and those of walls with bar reinforcement were greater by 17% (*ρ* = 0.05%) and 49% (*ρ* = 0.10%) when compared to unreinforced units. Figure 11b shows the comparison of shear strain angles in reinforced walls Θ*_cr,mv_* at the time of cracking and failure Θ*_u,mv_* obtained from tests on unreinforced units. Determined values of shear strain angles at the time of cracking were required for verifying SLS for structures. 

Figure 11b shows the comparison of angle values Θ*_cr, mv_* determined from tests on all unit series and the acceptable value equal to Θ*_adm_* = 0.5 mrad as specified in PN-B-03002:2007 [38] for unreinforced brick masonry to determine SLS. The diagram indicates that the limit value of an angle Θ*_adm_* was lower than that determined from tests on angle values Θ*_cr_* in all reinforced walls except for shear walls. Therefore, setting values Θ*_adm_* (specified in Reference [38]) for shear walls with reinforcement (infill walls) when no relevant regulations have been introduced can lead to dangerous underestimation of width of diagonal cracks.

#### 4.2.2. Walls Made of Calcium-Silicate Masonry Units

In walls made of silicate masonry units, the relationships *τ*-Θ (Figure 12) were very nonlinear from the beginning until the moment of failure. They were almost proportional until the moment of cracking. In models HOS-15/1, HOS-Z1-S-15/1, and HOS-Z2-S-15/1 under maximum compression, strengthening and further increase in shear–strain angle with an increasing shear loading were observed after cracking (slight breaking of the line on the diagram), and in shear walls without reinforcement, HOS-00/1 and HOS-010/1, and with reinforcement, HOS-Z1-S-010/1 and HOS-Z2-S-010/1, under initial compressive stress of 0 and 0.1 N/mm^2^, an increase in shear–strain angle with a small increase in shear loading (slightly inclined plate on diagrams) was observed. A further increase in horizontal loading resulted in slight strengthening of the wall. The test results are presented in Table 5.

At minimum compression of 6% and initial compressive stress of 1.5 N/mm^2^, cracking stresses in truss-reinforced units of series HOS-Z1-S were lower by 29% compared to unreinforced units. Failure stress increased at the time of failure by 12% in the wall under minimum compression and by 18% when initial compressive stresses were the highest; see Figure 13a. For models of series HOS-Z2-S reinforced with plastic mesh, cracking stresses in units under minimum compression only was only higher by 7% than in the same units without reinforcement. An increase in cracking stress in the unit under maximum compressive stress was lower by 6% when compared to the unreinforced model. Considering a failure stress increase with reference to unreinforced units, its increase was the greatest in the model under minimum compressive stress (by 21%) and lower by 2% in the unit under maximum compressive stress; see Figure 13a.

Shear strain in models of series HOS-Z1-S reinforced with plastic mesh, comparable to that in unreinforced units (Figure 13b), was found at the time of cracking in the unit under minimum compressive stress. However, the shear strain was lower even by 14% than in the unreinforced unit when the initial compressive stress of 1.5 N/mm^2^ was exerted. Shear deformation in units reinforced with truss were higher by more than 79% than in unreinforced units in the model under minimum compressive stress and lower by 10% than in the unit under maximum compressive and shear stress. 

At the time of cracking, shear strain in models of series HOS-Z2-S reinforced with plastic was greater by 27% in the unit under minimum compressive stress than in the unreinforced unit, but shear strain was lower even by 27% than in the unreinforced unit when the initial compressive stress of 1.5 N/mm^2^ was exerted. Shear deformation of units with plastic mesh reinforcement were greater by more than 38% in the model under minimum compressive and shear stress and lower by 48% in the unit under maximum compressive and shear stress; see Figure 13b. Figure 13b shows the comparison of test results with acceptable values of shear–strain angles equal to Θ*_adm_* = 0.4 mrad. Values of shear strain angle recommended by the standard [38] turned out to be a dangerous estimation for both unreinforced walls and walls with truss and plastic mesh reinforcement.

#### 4.2.3. Walls Made of AAC Masonry Units

Like in previously discussed results from tests on calcium silicate masonry units, also in walls made of AAC masonry units, relationships *τ*-Θ were nearly directly proportional until the time of destruction—Figure 14. For the model marked as HOS-AAC-075/1, compressed to the value of 0.75 N/mm^2^, no strengthening was observed after cracking (slight breaking of the line on the graph); only values of shear strain angle increased. In the unit HOS-AAC-010/1 subjected to minimum compression, some strengthening occurred after cracking—Figure 14a. Consequently, an increase in load also led to an increase in shear deformation angle. Noticeable strengthening with simultaneously smaller shear deformation was observed in the unit HOS-AAC-10/2 under maximum compression—Figure 14b. The test results are presented in Table 6.

Until the moment of cracking, the nature of the shear–stress angle of shear deformation relationship for walls reinforced with truss of series HOS-AAC-Z1-010/1, HOS-AAC-Z1-10/1, and with plastic mesh and of series HOS-AAC-Z2-010/1 and HOS-AAC-Z2-10/1 did not differ substantially from results obtained for unreinforced walls. Strengthening of all reinforced walls was noticed after cracking. After achieving the maximum shear strain, all elements revealed their plastic properties—values of shear strain angles increased without any increase in shear strain. The greatest deformations during that loading phase were exhibited by elements reinforced with truss. After achieving the maximum shear strain, deformations of elements reinforced with plastic mesh were smaller when compared to elements with Z1 reinforcement and greater than in case of unreinforced elements. When comparing units of series HOS-AAC-Z1 with truss reinforcement to unreinforced units, an increase in cracking stress equal to 17% was found only in the reinforced model under maximum compression, and in the model under minimum compression, values of cracking stress were lower by 3%; see Figure 15a. At the time of failure, shear stress in the model under minimum compression was 0.250 N/mm^2^, whereas shear stress in the compressed element did not exceed 0.50 N/mm^2^. With reference to similar unreinforced elements, the increases in stress values were 6% and 30%, respectively.

In HOS-AAC-Z2 walls reinforced with plastic mesh, the increases in cracking stress values were 5% and 13%, respectively, with reference to unreinforced elements. With reference to unreinforced elements, an increase in stress at the time of failure was found only in the model under maximum compression and was equal to 19%, and in the model under minimum compression, there was no increase in stress; see Figure 15a.

Regarding units of series HOS-AAC-Z1 with truss reinforcement, angles of shear deformation corresponding to failure stresses were wider by 54% and 74%, respectively, and at the time of cracking, the angle values achieved were 27% (0.1 N/mm^2^) and 33% (0.5 N/mm^2^); see Figure 15b. An increase in compressive prestress was accompanied by increasing shear strain and deformation angles in walls compressed to 0.5 N/mm^2^, and those values were 94% and 69%, respectively. Angles of shear strain and deformation read at the time of cracking and failure in the wall reinforced with plastic mesh under minimum initial compressive stresses were lower by 18% compared to strains in the unreinforced wall. The shear angle increased to 72% in the wall under maximum compressive stress; see Figure 15b. Figure 15b shows the comparison of test results with acceptable values of shear–strain angle equal to Θ*_adm_* = 0.4 mrad. Considering unreinforced walls, the recommended angle value proved to be the safe estimation for all walls except for the unit under minimum compressive stress. Reinforced units demonstrated a similar tendency. Assuming values specified in the standard [38] for walls under initial compressive stress proved to be a safe limitation.

## 5. Analysis on Effects of Reinforcement in Bed Joints

Table 7 presents results from our own tests on reinforced walls compared to results for unreinforced walls tested under the same values of initial compressive stress. Results other than shear–strain angle obtained at the time of cracking indicated an increase in average results for reinforced walls when compared to unreinforced models. The minimum quantity of used reinforcement was found to increase the value of analysed parameters. For shear stress, values increased by ca. 25–34% at the time of cracking and failure and that at the initial stiffness and the stiffness at the time of cracking increased by 70% and 58%, respectively. Also, the shear deformation angle increased by 7% at the time of failure and decreased by 11% on average at the time of cracking. Obtained results from our own tests were compared with tests performed by other authors. Due to the limited research material, the comparison only included stresses. 

### 5.1. Cracking and Ultimate Shear Stresses

Figure 16a compares the test results with reference to the relationship between the failure stress in reinforced walls *τ_u,z_* and the stress determined in unreinforced walls *τ_u,n_*. The achieved results were presented depending on the percentage of horizontal reinforcement *ρ* (values in brackets express the percentage of the horizontal reinforcement-*ρ*).

The above Table 7 indicates that the increase in reinforcement percentage did not cause a clear and proportional increase in the load capacity. It could be even argued that failure stress in reinforced walls was decreasing with an increasing quantity of reinforcement. In some tests conducted by Ernst [18], Ančić, Steinman [29], and Jasiński [39,40], the achieved values of load capacity were lower than in unreinforced walls. Furthermore, tests on walls made of concrete blocks with vertical and horizontal reinforcement, conducted by Haach, Vsconcelos, and Lourenço [23], demonstrated the highest increase in failure stress at the lowest reinforcement percentage, which could suggest that reinforcement percentage was not the only factor decisive for the load capacity of the wall. Figure 16b shows a comparison of the load capacity of all units only with the horizontal reinforcement as the function of compressive strength of the mortar *f*_m_. For mortars having *f_m_* < 3 N/mm^2^, the failure stress in the reinforced wall (*ρ* = 0.168–0.301%) made of hollow bricks (Ančić, Steinman [29]) was significantly lower than in the unreinforced wall. An increase in failure stress was observed with an increase with the mortar strength 3 N/mm^2^ < *f_m_* < 4.5 N/mm^2^ (units from solid brick at *ρ* = 0.146–0.267%) when compared to unreinforced units. Units made of hollow brick tested by Ančić, Steinman [29], which had the lowest reinforcement percentage *ρ* = 0.112%, were an exception. For walls made with mortar *f_m_* > 4.5 N/mm^2^, an increase in failure stress (at *ρ* = 0.168–0.187%) was definitely higher in the majority of tests, except for one model (*f_m_* = 6.0 N/mm^2^, *ρ_h_* = 0.079%) tested by Ernst [18] and in tests performed by Sanpaelesi and Cieni [28] (*f_m_* = 13.7 N/mm^2^, *ρ* = 0.187%), in which a slight increase was only observed. The smooth bar and truss-type reinforcement was found to have a positive effect on tested brick walls with the mortar *f_m_* = 9.67 N/mm^2^. A proportional increase in failure stress was observed with an increasing percentage of reinforcement. The most positive increase in the load capacity was found in tests conducted by Haach, Vasconcelos, and Lourenço [23], in which the cement mortar having the strength *f_m_* = 18.77 N/mm^2^ was used. Considering tests on calcium silicate made of silicate masonry units with the similar mortar strength *f_m_* = 18.2 N/mm^2^, an increase in failure stress was not significant when compared to unreinforced units. In tests on AAC masonry units [39,40] with the mortar strength of class M5, maximum values of failure stress were lower than in unreinforced walls. Exceptions were walls with truss reinforcement, in which the mortar was doubled on support areas of masonry units. The highest increase in stress, comparable to an increase in the strength of brick masonry, was achieved for the mortar of class M10 and truss-type reinforcement. 

A knowledge of cracking stress is a crucial issue describing the effect of reinforcement on the wall behaviour. Figure 17a presents a comparison of the test results expressed as the relationship between cracking stress in the reinforced wall and the unreinforced wall depending on the percentage content of the horizontal reinforcement (values in brackets express the percentage of the horizontal reinforcement *ρ*). For stress values determined at the time of cracking, the tendency was almost the same as for failure stress, that is, a drop-in cracking stresses was found with an increasing percentage of reinforcement when compared to unreinforced masonry. Considering cracking issues, the range of 0.1–0.2% was the least favourable, at which most of unreinforced units were cracked earlier than unreinforced models as found in tests performed by Ančić and Steinman [29]. An increase in cracking stress was found in every test on brick masonry. Only in tests on AAC masonry units [39,40], cracking stresses were slightly lower than in case of unreinforced walls (in case of one-side application of mortar). Proceeding in a similar manner as in case of the failure state, Figure 17b illustrates the results from tests on crack resistance as the function of compressive strength *f_m_* of mortar used in tested models. Those results were close to the ones achieved at failure. In the case of mortar with the strength *f_m_* < 3.5 N/mm^2^, values of cracking stress were considerably lower when compared to unreinforced walls (Ančić, Steinman [29])—solid and hollow brick (at *ρ* = 0.146–0.301%). Considering mortar with the strength 3.5 N/mm^2^ < *f_m_* < 4.5 N/mm^2^, ratios *τ_cr,z_*/*τ_cr,n_* were higher or lower than one for models made of solid brick and hollow brick (at *ρ* = 0.112–0.267%). An increase in cracking stress was observed only at *f_m_* > 5.0 N/mm^2^ and *ρ* = 0.125–0.187% (Ančić, Steinman [29]—solid and hollow brick, and Scrivener [17]—concrete units). An increase in cracking stress determined from tests on brick walls reinforced with bars was similar as in walls with the truss-type reinforcement and doubled mortar applied on bed joints in masonry units [39,40].

Due to the limited research material, it is difficult to draw far-reaching conclusions about the qualitative or quantitative aspects. Both the reinforcement percentage and the class of applied mortar seem to have a significant impact on crack-resistance and load capacity of the shear wall. However, mortar strength/adhesion to masonry units can be the decisive factor.

The tests also demonstrated the impact of reinforcement on the method of damage to the wall and its behaviour after cracking. In units with mixed reinforcement, the number of cracks at the time of failure was considerably higher than in units with horizontal reinforcement or without reinforcement, as in Ernst research [11]. Qualitative and quantitative changes were also observed in the wall after cracking. Strain in reinforced units was many times greater than in unreinforced models (Ernst [18], Scrivener [17], Jasiński [40], Haach, Vsconcelos, and Lourenço [23]), in which failure was rapid.

Figure 16 and Figure 17 show diagrams of the standard distribution of relationships *τ_u,z_*/*τ_u,n_* and *τ_cr,z_*/*τ_cr,n_*. There were 92 test results for failures stress, in which the average ratio between failure shear stress in reinforced walls and unreinforced walls was *τ_u,z_*/*τ_u,n_* = 1.373 and the corresponding standard deviation was *σ* = 0.468. The probability of reinforcement negative effect, determined on the above basis, was not greater than 21%. A similar analysis was performed for stress values at the time of cracking, and the available number of test results was 87. The average ratio between failure shear stress in reinforced walls and unreinforced walls was *τ_u,z_*/*τ_u,n_* = 1.363, and the corresponding standard deviation was *σ* = 0.692. The probability of a negative effect of the reinforcement was greater than at the time failure and was equal to 30%.

Boundary values in confidence intervals of the average value [41] (at n > 30 and unknown variance *σ*) expressing the reinforcement impact were determined form the general relationship at the statistical significance *α* = 0.8 in Equation (7):(7)$$P\left(\bar{x}-u_{1-\alpha /2}\frac{S}{\sqrt{n}}<m<\bar{x}+u_{1-\alpha /2}\frac{S}{\sqrt{n}}\right)=1-\alpha ,$$
where $\bar{x}$ is the mean value of the random sample, *S* is the standard deviation of the sample, and $u_{1-\alpha /2}$ is the statistics with the random variable with the standard distribution N(0.1).

The following values were obtained for the analysed issues:

Cracking stresses *τ_u,z_*/*τ_u,n_* in Equation (8):(8)$$1.363-1.945\frac{0.692}{\sqrt{87}}<\tau_{cr,z}/\tau_{cr,n}<1.363+1.945\frac{0.692}{\sqrt{87}}\to 1.218<\tau_{cr,z}/\tau_{cr,n}<1.507,$$

Failure stresses *τ_u,z_*/*τ_u,n_* in Equation (9): (9)$$1.373-1.766\frac{0.468}{\sqrt{92}}<\tau_{u,z}/\tau_{u,n}<1.373+1.766\frac{0.468}{\sqrt{92}}\to 1.278<\tau_{u,z}/\tau_{u,n}<1.467.$$

Regarding the probability up to 20%, reinforcement in bed joints seems to increase the average values of cracking stress by 23% and the average values of failure stress by 28%.

### 5.2. Shear Strain and Stiffness

As there are no reliable test results, a similar analysis cannot be performed for achieved values of shear strain and stiffness. Shear strain angle and shear deformation ratios were obtained from our own results. These values are shown in Table 7. As the sample size was small, *n* < 30, the Equation (10) was used:(10)$$P\left(\bar{x}-t_{1-\alpha /2}\frac{S}{\sqrt{n}}<m<\bar{x}+t_{1-\alpha /2}\frac{S}{\sqrt{n}}\right)=1-\alpha ,$$
where $\bar{x}$ is the mean value of the random sample, *S* is the standard deviation of the sample, and $t_{1-\alpha /2}$ is the statistics with the Student’s t-distribution and n−1 degrees of freedom. 

Regarding shear–strain angles at the time of cracking, the following Equation (11) was obtained:(11)$$0.89-1.319\frac{0.244}{\sqrt{24}}<\Theta_{cr,z}/\Theta_{cr,n}<0.89+1.319\frac{0.244}{\sqrt{24}}\to 0.85<\Theta_{cr,z}/\Theta_{cr,n}<0.92.$$

For shear deformation angle, the following Equation (12) was obtained: (12)$$1.07-1.319\frac{0.419}{\sqrt{24}}<\Theta_{u,z}/\Theta_{u,n}<1.07+1.319\frac{0.419}{\sqrt{24}}\to 0.95<\Theta_{u,z}/\Theta_{u,n}<1.18.$$

Like in the case of stress values, reinforcement in bed joints—regarding the probability up to 20%—seems to reduce average values of shear strain angle by 8% at the time of cracking and, simultaneously, to increase average values of shear deformation angle by 18%.

Analyses performed in a similar way as described above on initial values of wall stiffness *K_o_* and stiffness at the time of cracking *K_cr_* were used to obtain the in Equations (13) and (14):(13)$$1.70-1.319\frac{0.648}{\sqrt{24}}<K_{o,z}/K_{o,n}<1.70+1.319\frac{0.648}{\sqrt{24}}\to 1.52<K_{o,z}/K_{o,n}<1.87,$$
(14)$$1.58-1.319\frac{0.801}{\sqrt{24}}<K_{cr,z}/K_{cr,n}<1.58+1.319\frac{0.801}{\sqrt{24}}\to 1.36<K_{cr,z}/K_{cr,n}<1.79.$$

On the basis of defined limits of confidence intervals, reinforcement in bed joints is found at the probability not exceeding 20% to increase average values of initial stiffness by 52% and the average stiffness at the time of cracking by 36%.

## 6. Conclusions

Our own tests and comparative analyses of other tests are sufficient to come to the following conclusions regarding the effect of reinforcements used in bed joints in brick masonry made of calcium silicate and AAC units: 

1) Observations of failure methods of shear unreinforced and reinforced walls indicate the following:
Initial compressive stress was the factor affecting crack morphology. In walls subjected to minimum compressive stress, there was a predominant single crack running through head and bed joints, whereas in walls subjected to maximum compressive stress, including masonry units, there were many diagonal and even vertical cracks;horizontal reinforcement in bed joints constrained the number of cracks;differences in masonry behaviour were observed at the phase close to failure as unreinforced units or the ones with plastic mesh type reinforcement were gently wearing out, and masonry with truss type reinforcement were destroyed immediately by crushing with simultaneous reinforcement breaking.

2) Regarding shear stress at the time of cracking *τ_c_* and failure *τ_u_*, the following observations were made:the noticeable effect of compressive stress on values of shear stress at the time of cracking and failure was confirmed;steel reinforcement in the form of unbonded steel bars and trusses used in the minimum quantity in solid brick walls (acc. to PN-EN 1996-1-1 [32]) *ρ*_min_ = 0.1%, and lower than minimum quantity did not result in an undesirable reduction of shear stress at the time of cracking and failure;the average increases in cracking and failure stress were 25% and 34%, respectively;the conducted statistical analysis of our own tests and those by other authors indicated that the reinforcement placed in bed joints increases average values of cracking and failure stress by 22% and 28%, respectively.

3) Regarding shear–strain angles at the time of cracking Θ*_cr_* and failure Θ*_u_*, the following observations were made:a significant impact of initial compressive stress was found in all tested series of units, and the tendency was that an increase in initial compressive stress resulted in increased angles of shear strain;generally, at the time of cracking, reinforcement reduced angles of shear strain by 11% on average and increased angles of shear deformation by 7% on average;including statistical analyses, shear–strain angles decreased by 8%, and an increase of shear deformation was equal to 18%,limitations of shear–strain angle, accepted in Polish design rules PN-B-03002:2007 [38], which meet SLS conditions, were found to be dangerous for unreinforced and reinforced walls made of solid brick and AAC and evidently overestimated for Ca-Si walls.

4) Considering the initial stiffness *K_o_* and stiffness at the time of cracking *K_cr_*, it was the following were found:the highest increase in the initial stiffness and stiffness at the time of cracking was observed in walls under maximum compression;in reinforced walls, there was a noticeable increase in the initial stiffness *K_o_* and stiffness at the time of cracking *K_cr_* by 70% and 58% on average;after taking into account statistical analyses, reinforcement in bed joints caused an increase in average values of *K_o_* and *K_cr_* by 52% and 36%.

## Figures and Tables

**Figure 1 materials-12-02543-f001:**
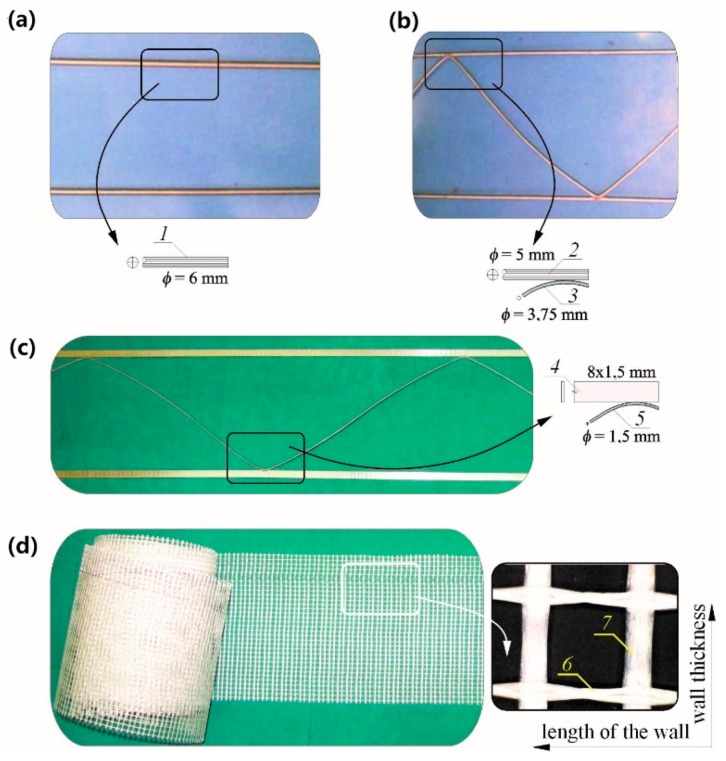
Reinforcements used in the tests: (**a**,**b**) reinforcement in solid brick walls and (**c**,**d**) reinforcement in a wall made of Ca-Si and autoclaved aerated concrete (AAC) masonry units; 1—stainless steel bars, 2—truss strips made of bars with a diameter of 5 mm, 3—truss struts made of bars with a diameter of 3.75 mm, 4—truss strips made of 8 × 1.5 mm flat bars, 5—truss struts with a diameter of 1.5 mm, 6—weft fibres, and 7—warp fibres.

**Figure 2 materials-12-02543-f002:**
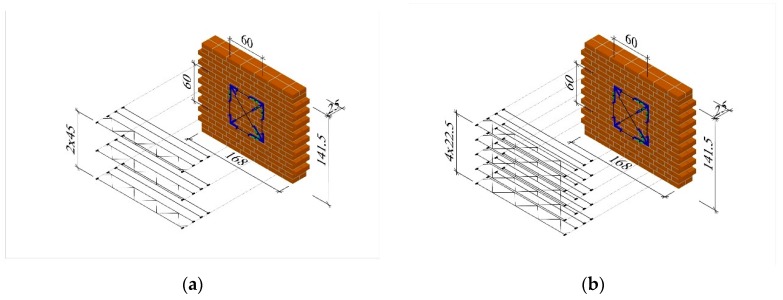
Geometry of solid brick models: (**a**) reinforced with bars and trusses of series HC-ZPI and HC-ZKI (*ρ* = 0.05%) and (**b**) reinforced with bars and trusses of series HC-ZPII and HC-ZKII (*ρ* = 0.10%), dimensions are in centimeters.

**Figure 3 materials-12-02543-f003:**
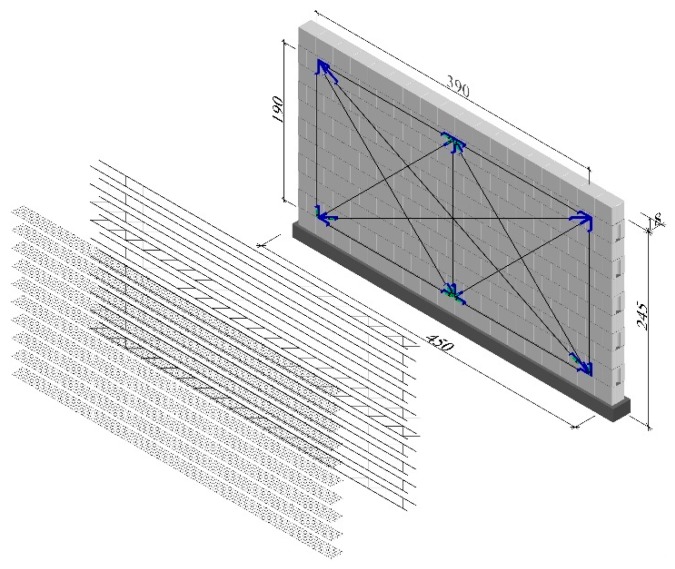
Geometry of models made of silicate masonry units, dimensions are in centimeters.

**Figure 4 materials-12-02543-f004:**
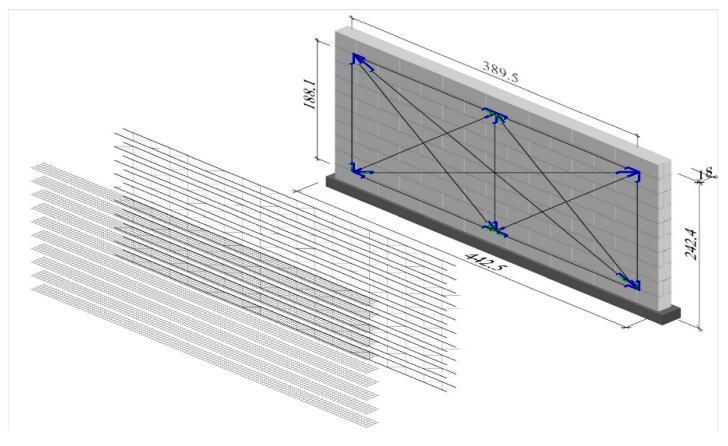
Geometry of models made of AAC masonry units, dimensions are in centimeters.

**Figure 5 materials-12-02543-f005:**
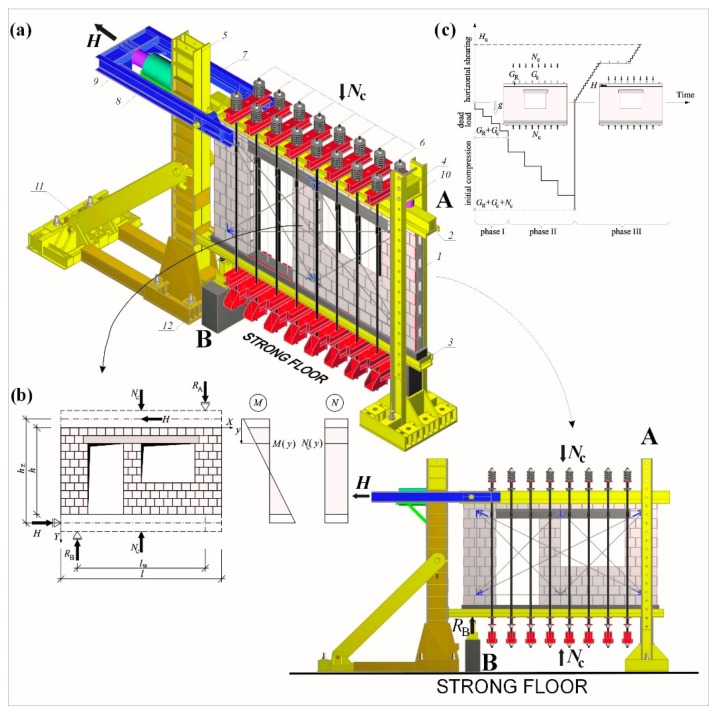
(**a**) Test stand, (**b**) static scheme, and (**c**) sequences of loads.

**Figure 6 materials-12-02543-f006:**
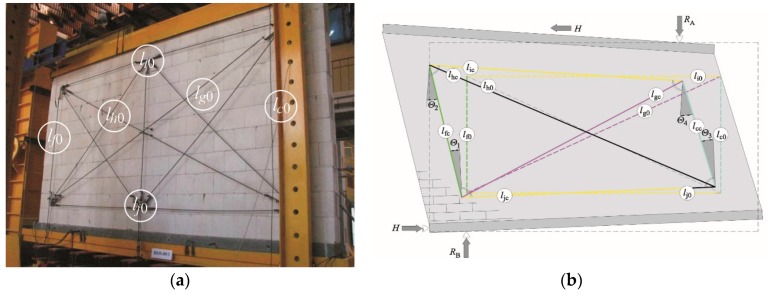
Frame system for measuring strain and deformation angle: (**a**) a wall made of calcium silicate units and (**b**) determining measurement bases and partial strain angles.

**Figure 7 materials-12-02543-f007:**
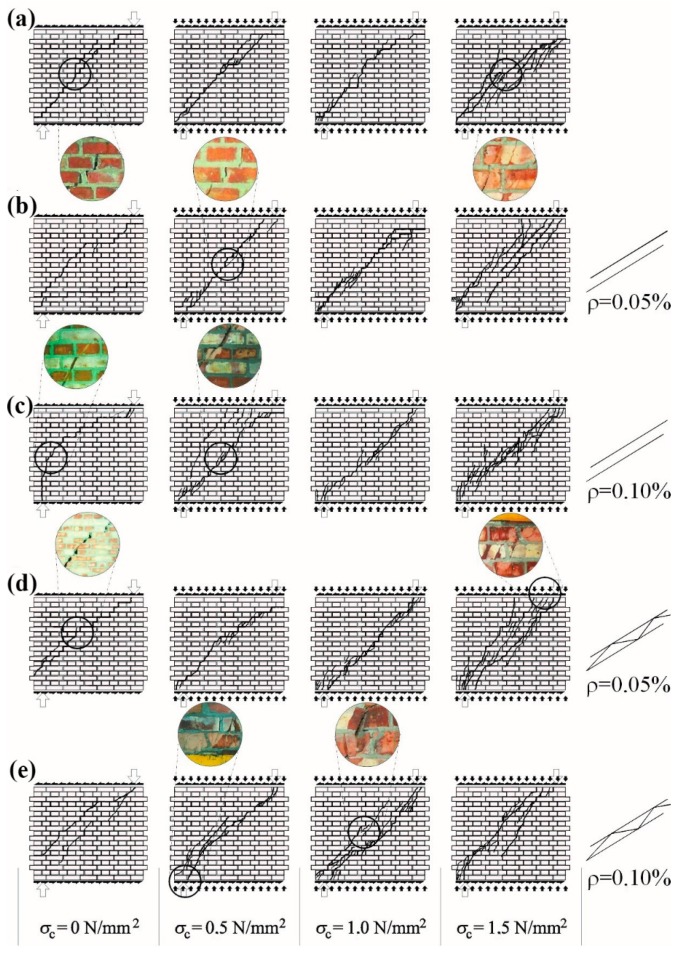
Patterns of cracking clay brick masonry walls: (**a**) unreinforced walls, (**b**) walls reinforced with bars *ρ* = 0.05%, (**c**) walls reinforced with bars *ρ* = 0.10%, (**d**) walls reinforced with trusses *ρ* = 0.05%, and (**e**) walls reinforced with trusses *ρ* = 0.10%.

**Figure 8 materials-12-02543-f008:**
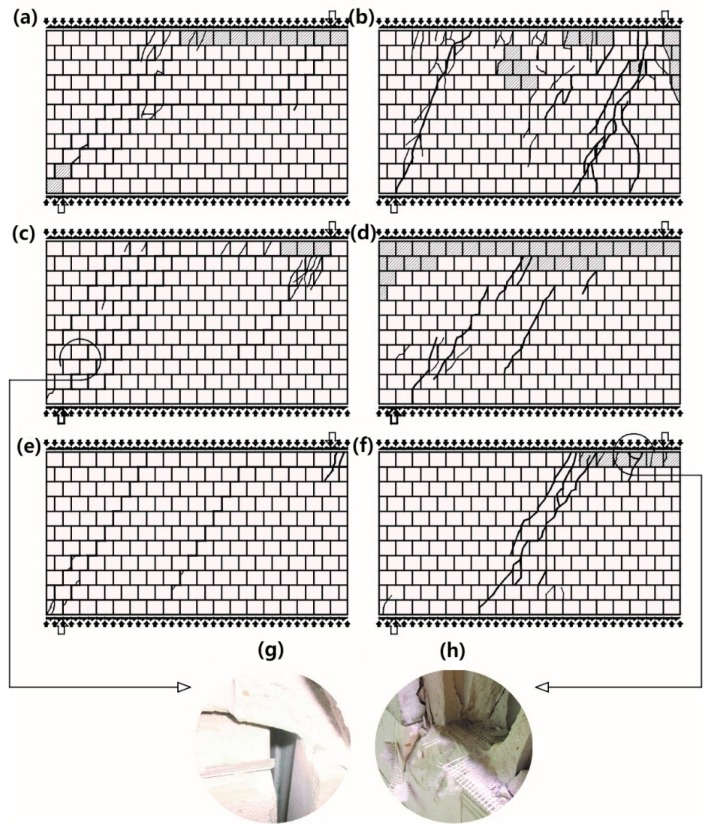
Cracking patterns of HOS series walls at the time of failure: (**a**) unreinforced shear wall at *σ_c_* = 0.1 N/mm^2^, (**b**) unreinforced shear wall at *σ_c_* = 1.5 N/mm^2^, (**c**) shear wall reinforced with steel trusses at *σ_c_* = 0.1 N/mm^2^, (**d**) shear wall reinforced with steel trusses at *σ_c_* = 1.5 N/mm^2^, (**e**) shear wall reinforced with plastic mesh at *σ_c_* = 0.1 N/mm^2^, (**f**) shear wall reinforced with plastic mesh at *σ_c_* = 1.5 N/mm^2^, (**g**) a broken truss, and (**h**) a broken plastic grid in the crush zone of a wall.

**Figure 9 materials-12-02543-f009:**
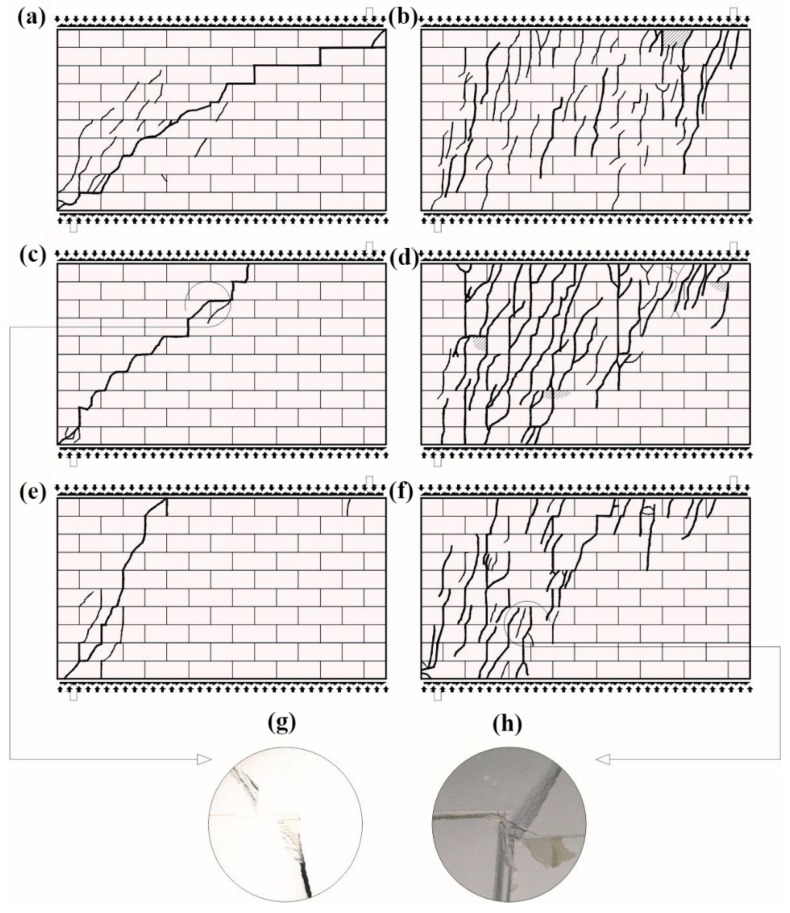
Cracking patterns of HOS-AAC series walls at the time of failure: (**a**) unreinforced shear wall at *σ_c_* = 0.1 N/mm^2^, (**b**) unreinforced shear wall at *σ_c_* = 1.0 N/mm^2^, (**c**) shear wall reinforced with steel trusses at *σ_c_* = 0.1 N/mm^2^, (**d**) shear wall reinforced with steel trusses at *σ_c_* = 1.0 N/mm^2^, (**e**) shear wall reinforced with plastic mesh at *σ_c_* = 0.1 N/mm^2^, (**f**) shear wall reinforced with plastic mesh at *σ_c_* = 1.0 N/mm^2^, (**g**) a broken truss, and (**h**) a broken plastic grid in the crush zone of a wall.

**Figure 10 materials-12-02543-f010:**
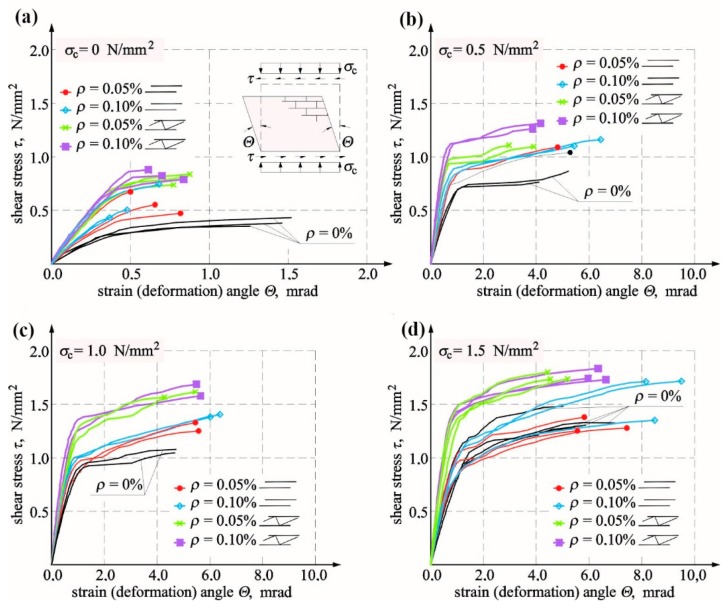
Relationships *τ* − Θ for unreinforced and reinforced units made of solid brick tested at different values of initial compressive stress: (**a**) *σ_c_* = 0, (**b**) *σ_c_* = 0.5 N/mm^2^, (**c**) *σ_c_* = 1.0 N/mm^2^, and (**d**) *σ_c_* = 1.5 N/mm^2^.

**Figure 11 materials-12-02543-f011:**
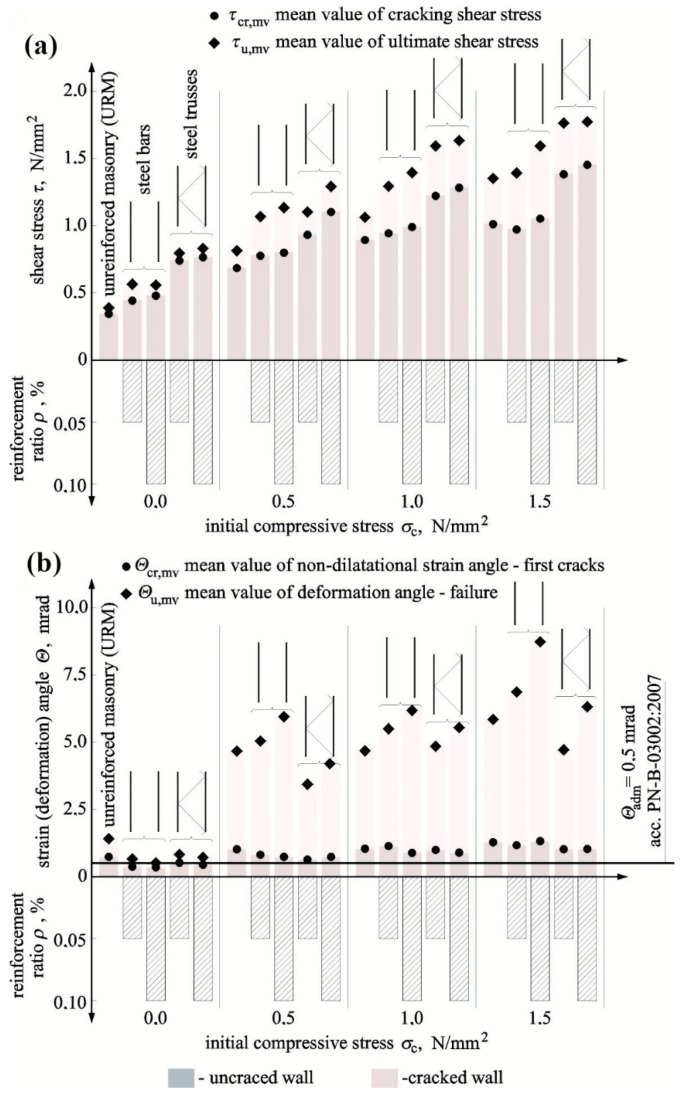
Comparison of test results: (**a**) shear stress at the time of cracking and failure and (**b**) shear–strain angle at the time of cracking and shear deformation angle at the time of failure.

**Figure 12 materials-12-02543-f012:**
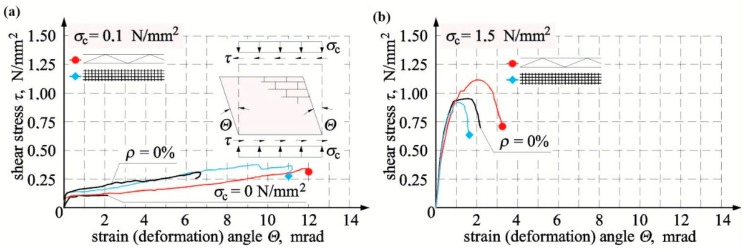
Relationships *τ*-Θ for unreinforced and reinforced units made of silicate masonry units tested at different values of initial compressive stress: (**a**) *σ_c_* = 0.1 N/mm^2^ and (**b**) *σ_c_* = 1.5 N/mm^2^.

**Figure 13 materials-12-02543-f013:**
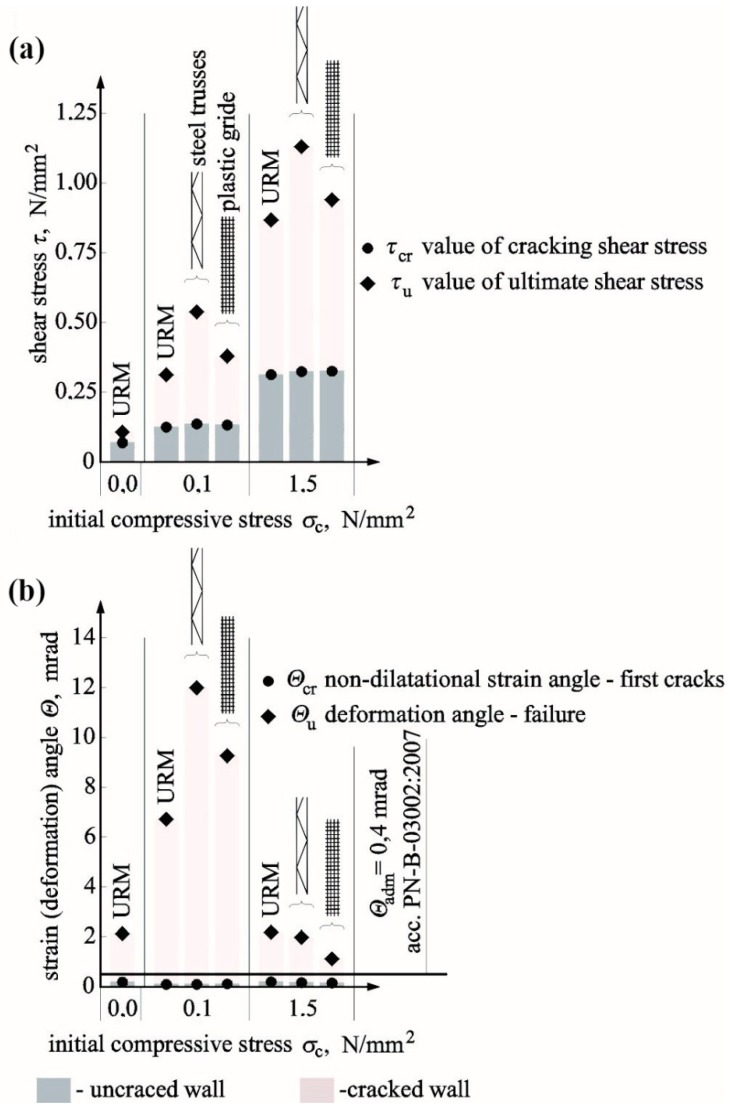
Comparison of test results: (**a**) shear stress at the time of cracking and failure and (**b**) shear–strain angle at the time of cracking and shear deformation angle at the time of failure.

**Figure 14 materials-12-02543-f014:**
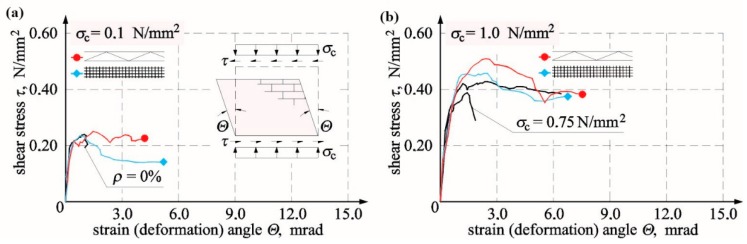
Relationships *τ_v,i_-*Θ*_i_* for unreinforced and reinforced AAC masonry tested at different values of initial compressive stress: (**a**) *σ_c_* = 0.1 N/mm^2^ and (**b**) *σ_c_* = 1.5 N/mm^2^.

**Figure 15 materials-12-02543-f015:**
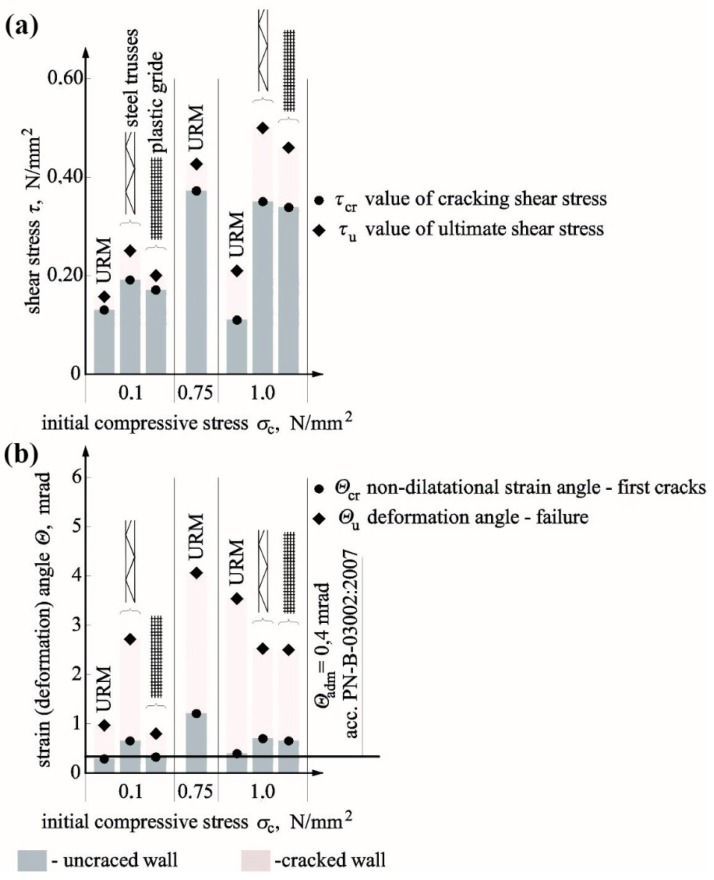
Comparison of test results: (**a**) shear stress at the time of cracking and failure and (**b**) shear–strain angle at the time of cracking and shear deformation angle at the time of failure.

**Figure 16 materials-12-02543-f016:**
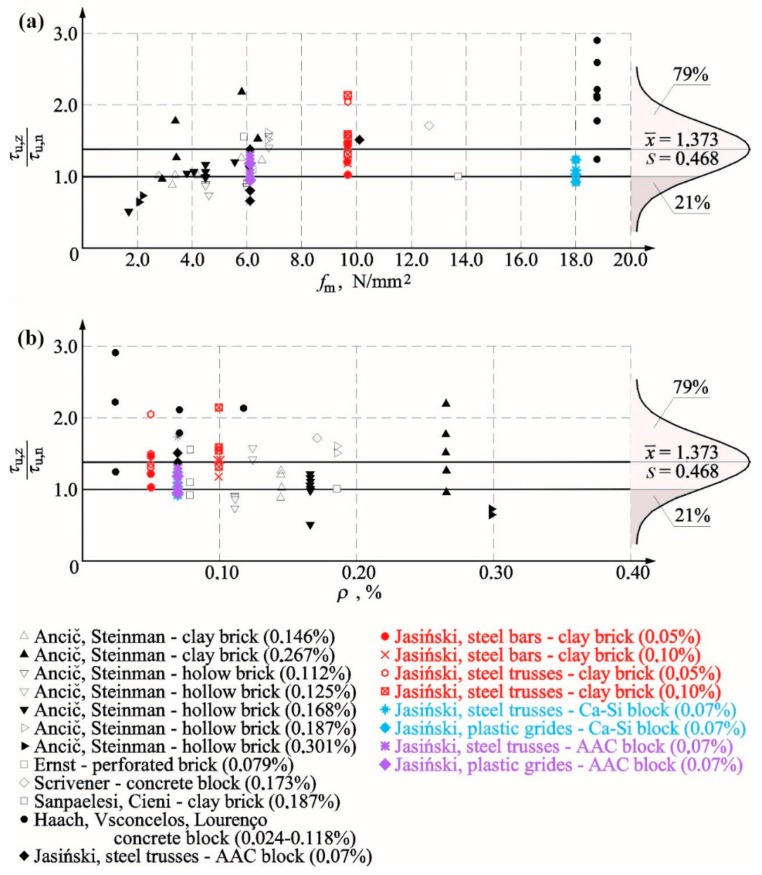
Summary of test results: (**a**) failure shear stress value for reinforced wall *τ_u,z_*/failure shear stress value for unreinforced wall *τ_u,n_*—depending on percentage of horizontal reinforcement *ρ*—and (**b**) ratio *τ_u,z_*/*τ_u,n_—*compressive strength of mortar *f_m_*.

**Figure 17 materials-12-02543-f017:**
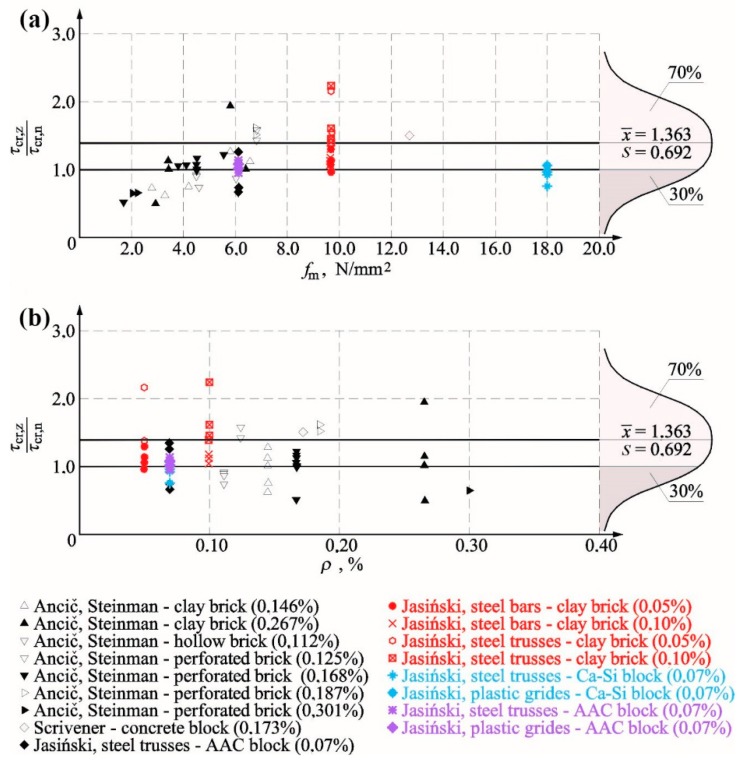
Summary of test results: (**a**) cracking stress value for reinforced wall *τ_cr,z_*/cracking shear stress value for unreinforced wall *τ_cr,n_*—depending on the percentage of horizontal reinforcement *ρ*—and (**b**) ratio *τ_cr,z_*/*τ_cr,n_—*compressive strength of mortar *f_m_*_._

**Table 1 materials-12-02543-t001:** Research programme for brick walls.

Series Marking	Wall Dimensions *h/l*, m	Type of Reinforcement	Reinforcement % *ρ*, %	*σ_c_* N/mm^2^	Number of Test Units	
					at *σ_c_*	Total
HC	1.42/1.68	Without reinforcement	-	0	3	11
				0.5	2	
				1.0	2	
				1.5	4	
HC-ZPI		Smooth bars *ϕ* 6 mm (Figure 1a)	0.05	0	3	10
				0.5	2	
				1.0	2	
				1.5	3	
HC-ZPII		Smooth bars *ϕ* 6 mm (Figure 1a)	0.10	0	3	10
				0.5	2	
				1.0	2	
				1.5	3	
HC-ZKI		Trusses (Figure 1b)	0.05	0	3	10
				0.5	2	
				1.0	2	
				1.5	3	
HC-ZKII		Trusses (Figure 1b)	0.10	0	3	10
				0.5	2	
				1.0	2	
				1.5	3	

**Table 2 materials-12-02543-t002:** Research programme for walls made of calcium silicate masonry units.

Series Marking	Wall External Dimensions *h/l*, m	Type of Reinforcement	Reinforcement % *ρ*, %	*σ_c_* (N/mm^2^)	Number of Test Units	
					at *σ_c_*	Total
HOS	2.45/4.50	Without reinforcement	0	0	1	3
				0.1	1	
				1.5	1	
HOS-Z1-S		Trusses (Figure 1c)	0.07	0.1	1	2
				1.5	1	
HOS-Z2-S		Plastic meshes (Figure 1d)	0.07	0.1	1	2
				1.5	1	

**Table 3 materials-12-02543-t003:** Research programme for walls made of AAC masonry units.

Series Marking	Wall External Dimensions *h/l*, m	Type of Reinforcement	Reinforcement % *ρ*, %	*σ_c_* (N/mm^2^)	Number of Test Units	
					at *σ_c_*	Total
HOS-AAC	2.43/4.43	Without reinforcement	0	0.1	1	4
				0.75	1	
				1.0	2	
HOS-AAC-Z1		Trusses (Figure 1c)	0.07	0.1	1	2
				1.0	1	
HOS-AAC-Z2		Plastic meshes (Figure 1d)	0.07	0.1	1	2
				1.0	1	

**Table 4 materials-12-02543-t004:** Test results for solid brick walls.

Type of Reinforcement	*ρ*,%	*σ_c_* N/mm^2^	Stresses		Angles of Shear Strain (Deformation)		Total Stiffness	
			Cracking	Failure	Cracking	Failure	Initial	At the Time of Cracking
			*τ_cr,mv_* N/mm^2^	*τ_u,mv_* N/mm^2^	Θ*_cr,mv_* mrad	Θ*_u,mv_* mrad	*K_o_*_, *mv*_ MN/m	*K_cr,mv_* MN/m
no reinforcement	0	0	0.343	0.388	0.735	1.413	301	118
		0.5	0.684	0.812	1.02	4.665	282	168
		1.0	0.892	1.06	1.04	4.671	374	214
		1.5	1.01	1.35	1.28	5.84	370	197
smooth rebars	0.05	0	0.442	0.564	0.373	0.658	577	305
		0.5	0.775	1.066	0.816	5.04	668	239
		1.0	0.942	1.291	1.14	5.49	605	206
		1.5	0.970	1.39	1.17	6.86	484	209
	0.1	0	0.479	0.557	0.347	0.510	493	346
		0.5	0.798	1.132	0.739	5.94	539	273
		1.0	0.988	1.392	0.888	6.17	624	264
		1.5	1.05	1.59	1.32	8.72	453	199
truss	0.05	0.0	0.739	0.794	0.523	0.827	732	353
		0.5	0.930	1.10	0.638	3.43	700	364
		1.0	1.22	1.59	0.994	4.84	593	308
		1.5	1.38	1.76	1.02	4.71	751	340
	0.1	0.0	0.764	0.829	0.445	0.717	740	430
		0.5	1.10	1.29	0.735	4.01	816	375
		1.0	1.28	1.63	0.892	5.54	717	357
		1.5	1.45	1.77	1.03	6.31	1095	353

**Table 5 materials-12-02543-t005:** Test results for walls made of silicate masonry units.

Type of Reinforcement	*ρ*,%	*σ_c_* N/mm^2^	Stresses		Angles of Shear Strain (Deformation)		Total Stiffness	
			Cracking	Failure	Cracking	Failure	Initial	At the Time of Cracking
			*τ_cr_* N/mm^2^	*τ_u_* N/mm^2^	Θ*_cr_* mrad	Θ*_u_* mrad	*K_o_* MN/m	*K_cr_* MN/m
no reinforcement	0	0	0.069	0.107	0.175	2.126	137	131
		0.1	0.124	0.313	0.086	6.714	1378	477
		1.5	0.346	0.954	0.197	2.182	1674	580
truss	0.07	0.1	0.088	0.35	0.087	11.99	1039	333
		1.5	0.324	1.13	0.169	1.968	1525	635
plastic mesh	0.07	0.1	0.133	0.379	0.109	9.262	1478	403
		1.5	0.326	0.939	0.143	1.125	1496	753

**Table 6 materials-12-02543-t006:** Test results for walls made of AAC masonry units.

Type of Reinforcement	*ρ*,%	*σ_c_* N/mm^2^	Stresses		Angles of Shear Strain (Deformation)		Total Stiffness	
			Cracking	Failure	Cracking	Failure	Initial	At the Time of Cracking
			*τ_cr_* N/mm^2^	*τ_u_* N/mm^2^	Θ*_cr_* mrad	Θ*_u_* mrad	*K_o_* MN/m	*K_cr_* MN/m
no reinforcement	0	0.1	0.196	0.235	0.281	0.97	932	229
		0.75	0.372	0.426	0.724	2.44	1168	169
		1.0	0.298	0.385	0.524	1.45	1541	187
		1.0 *	0.11	0.25	0.651	2.72	379	75
truss	0.07	0.1	0.191	0.250	0.358	1.49	1262	175
		1.0	0.350	0.50	0.695	2.52	1782	165
plastic mesh	0.07	0.1	0.205	0.23	0.322	0.80	1193	208
		1.0	0.338	0.46	0.649	2.50	1374	171

* An element neglected in further analyses.

**Table 7 materials-12-02543-t007:** Comparison of test results for reinforced walls of all tested series of units.

Wall Type	Type of Reinforcement	*ρ*,%	*σ_c_* N/mm^2^	Stresses		Angles of Shear Strain (Deformation)		Total Stiffness	
				Cracking	Failure	Cracking	Failure	Initial	At the Time of Cracking
				$\frac{\tau_{\mathrm{cr},z}}{\tau_{\mathrm{cr},n}}$	$\frac{\tau_{u,z}}{\tau_{u,n}}$	$\frac{\Theta_{\mathrm{cr},z}}{\Theta_{\mathrm{cr},n}}$	$\frac{\Theta_{u,z}}{\Theta_{u,n}}$	$\frac{K_{o,z}}{K_{o,n}}$	$\frac{K_{\mathrm{cr},z}}{K_{\mathrm{cr},n}}$
solid brick	smooth bars	0.05	0	1.29	1.45	0.51	0.47	1.92	2.58
			0.5	1.13	1.31	0.80	1.08	2.37	1.42
			1.0	1.06	1.22	1.10	1.18	1.62	0.96
			1.5	0.96	1.03	0.91	1.17	1.31	1.06
		0.1	0	1.40	1.44	0.47	0.36	1.64	2.93
			0.5	1.17	1.39	0.72	1.27	1.91	1.63
			1.0	1.11	1.31	0.85	1.32	1.67	1.23
			1.5	1.04	1.18	1.03	1.49	1.22	1.01
	truss	0.05	0.0	2.15	2.05	0.71	0.59	2.43	2.99
			0.5	1.36	1.35	0.63	0.74	2.48	2.17
			1.0	1.37	1.50	0.96	1.04	1.59	1.44
			1.5	1.37	1.30	0.80	0.81	2.03	1.73
		0.1	0.0	2.23	2.14	0.61	0.51	2.46	3.64
			0.5	1.61	1.59	0.72	0.86	2.89	2.23
			1.0	1.43	1.54	0.86	1.19	1.92	1.67
			1.5	1.44	1.31	0.80	1.08	2.96	1.79
wall made of silicate masonry units	truss	0.07	0.1	0.71	1.12	1.01	1.79	0.75	0.70
			1.5	0.94	1.18	0.86	0.90	0.91	1.09
	plastic mesh	0.07	0.1	1.07	1.21	1.27	1.38	1.07	0.84
			1.5	0.94	0.98	0.73	0.52	0.89	1.30
wall made of AAC masonry units	truss	0.07	0.1	0.97	1.06	1.27	1.54	1.35	0.76
			1.0	1.17	1.30	1.33	1.74	1.16	0.88
	plastic mesh	0.07	0.1	1.05	0.98	1.15	0.82	1.28	0.91
			1.0	1.13	1.19	1.24	1.72	0.89	0.91
Average value $\bar{x}$:				1.25	1.34	0.89	1.07	1.70	1.58
Standard deviation *S*:				0.355	0.287	0.244	0.419	0.648	0.801

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
