# Peer review of "Research on the Influence of Bed Joint Reinforcement on Strength and Deformability of Masonry Shear Walls"

_materials, 2019, doi:10.3390/ma12162543_

Round 1
Reviewer 1 Report
This paper (Research of influences of bed joints reinforcements on strength and deformability of masonry shear walls) presents interesting results but needs a thorough revision before being considered for publication. Some sections need to be rewritten, like the literature review and Conclusions.
Literature review: I found a number of papers on this topic that you did not cite. I am listing them here, please consider them in the literature review and in the interpretation of the results:
1. Estimation of in-plane shear capacity of confined masonry walls with and without openings using strut-and-tie analysis, Engineering Structures, Volume 188, 1 June 2019, Pages 290-304, D. Tripathy, V. Singhal
2. Assessing the performance of CFRP strengthening on masonry walls using experimental modal analysis, Engineering Structures, Volume 193, 15 August 2019, Pages 184-193, Ernest Bernat-Masó, Lluís Gil
3. Shear response of partially-grouted reinforced masonry walls with a central opening: Testing and detailed micro-modelling, Materials & Design, Volume 118, 15 March 2017, Pages 122-137, Sebastián Calderón, Cristián Sandoval, Oriol Arnau
4. Enhancing the out-of-plane performance of masonry walls using engineered cementitious composite, Composites Part B: Engineering, Volume 140, 1 May 2018, Pages 108-122, S. Pourfalah, B. Suryanto, D. M. Cotsovos
5. Modelling the out-of-plane behaviour of masonry walls retrofitted with engineered cementitious composites, Computers & Structures, Volume 201, May 2018, Pages 58-79, S. Pourfalah, D. M. Cotsovos, B. Suryanto
6. A study of the out-of-plane performance of brick veneer wall systems in medium rise buildings under seismic loads, Engineering Structures, Volume 48, March 2013, Pages 683-694, Niranjan Desai, W. M. McGinley
7. Seismic strengthening of infill walls with perforated steel plates, Engineering Structures, Volume 152, 1 December 2017, Pages 168-179, Bengi Aykaç, Eray Özbek, Rahim Babayani, Mehmet Baran, Sabahattin Aykaç
8. An investigation into the time dependency of shear strength of clay brick walls; an approximate approach, Construction and Building Materials, Volume 155, 30 November 2017, Pages 88-102, Mohammad Amir Sherafati, Mohammad Reza Sohrabi
9. FRCM strengthening of clay brick walls for out of plane loads, Composites Part B: Engineering, Volume 174, 1 October 2019, Article 107050, Claudio D'Ambra, Gian Piero Lignola, Andrea Prota, Francesco Fabbrocino, Elio Sacco
10. Experimental performance of FRCM retrofit on out-of-plane behaviour of clay brick walls, Composites Part B: Engineering, Volume 148, 1 September 2018, Pages 198-206, Claudio D'Ambra, Gian Piero Lignola, Andrea Prota, Elio Sacco, Francesco Fabbrocino
11. Experimental study on scale effects in clay brick masonry prisms and wall panels investigating compression and shear related properties, Construction and Building Materials, Volume 163, 28 February 2018, Pages 706-713, C. L. Knox, D. Dizhur, J. M. Ingham
12. Seismic behavior and improvement of autoclaved aerated concrete infill walls, Engineering Structures, Volume 193, 15 August 2019, Pages 68-81, Baris Binici, Erdem Canbay, Alper Aldemir, Ismail Ozan Demirel, Ahmet Yakut
13. Towards sustainable bricks production: An overview, Construction and Building Materials, Volume 165, 20 March 2018, Pages 112-125, Anant L. Murmu, A. Patel
Results and discussion: The paper presents a big amount of interesting results from unusual experiments. The analysis of the results is very good.
Conclusions: The discussion about technological benefit have to be separated in the article according points of conclusions.
Author Response
The answer to No. 1 REVIEW
Thank you very much, for a thorough review and finding interesting and valuable publications. Of course, some of the articles were familiar to me and I put in a review of literature. I have not used the analytical part in my article because I analyzed:
a) the walls do not have openings, b) walls loaded monotonically and not cyclically or dynamically in the plane, c) reinforcement was used only in bed joints (walls with reinforcement in vertical drillings were omitted), d) the authors provided the most important data on wall properties (compressive strength of mortar and masonry elements, reinforcement ratio, value of initial compressive stresses, etc.).After analyzing the content of the publication, I inform that:
Item 1: The publication concerns boundary walls with openings. The subject matter in the publication goes beyond the scope of my article,
Item 2: Work refers to wall reinforcement, loaded dynamically. The subject matter in the publication goes beyond the scope of my article,
Item 3: Work concerns walls with openings and filled with vertical drillings in masonry elements. The subject matter in the publication goes beyond the scope of my article,
Item 4: Work refers to walls loaded from the plane. The subject matter in the publication goes beyond the scope of my article,
Item 5: Work refers to walls loaded from the plane. The subject matter in the publication goes beyond the scope of my article,
Item 6: Work refers to diaphragm walls loaded from the plane. The subject matter in the publication goes beyond the scope of my article,
Item 7: Work applies to reinforced walls with seismic loads. It goes beyond the scope of my publication,
Item 8: Work concerns non-reinforced walls. It goes beyond the scope of my publication,
Item 9: Work refers to reinforced walls loaded from the plane. The subject matter in the publication goes beyond the scope of my article,
Item 10: Work refers to reinforced walls loaded from the plane. The subject matter in the publication goes beyond the scope of my article,
Item 11: Work does not apply to reinforced walls. The subject matter in the publication goes beyond the scope of my article,
Item 12: The work concerns walls filling skeletal costumes. The subject matter in the publication goes beyond the scope of my article,
Pos.13: The research results of reinforced walls are not presented in the paper. The subject matter in the publication goes beyond the scope of my article,
I also arranged the analysis of the research results to match the formulated conclusions.
Reviewer 2 Report
- Suggestion for minor change in title:
"Research on the influence of bed joint reinforcement on strength and ... ".
- Fig. 1, font for text/numbers could be larger.
- minor text/format corrections in lines; 17, 176, 388, 643 ("angels"), 652.
Author Response
The answer to No. 2 REVIEW
Thank you very much for the reviews. I have made appropriate corrections to the title. I also corrected obvious language errors.
Reviewer 3 Report
Strength and deformation Influence of reinforcement used in bed joints in brick masonry is important
to Polish markets. This work shows some important discoveries.
Author Response
The answer to No. 3 REVIEW
Thank you very much for the reviews. I introduced significant changes suggested by reviewers.